# Antibacterial Activity of Ferulic Acid Ester against *Ralstonia solanacearum* and Its Synergy with Essential Oils

Qing-Bo Tu [1,2], Hui-Cong Shi [1], Ping Li [1,3], Sheng Sheng [1,3] and Fu-An Wu [1,3,*]

1   School of Biotechnology, Jiangsu University of Science and Technology, Zhenjiang 212018, China
2   School of Pharmacy, Nanjing University of Chinese Medicine Hanlin College, Taizhou 225300, China
3   Key Laboratory of Silkworm and Mulberry Genetic Improvement, Ministry of Agricultural and Rural Affairs, Sericultural Research Institute, Chinese Academy of Agricultural Sciences, Zhenjiang 212018, China
*   Correspondence: fuword@163.com; Tel.: +86-511-85616571

**Abstract:** *Ralstonia solanacearum* is one of the ten most harmful plant bacteria worldwide, and traditional agrochemicals are not very effective in controlling this pathogen. Moreover, excessive pesticides always bring organic residues and resistant strains, which cause the unsustainability of the environment. In this paper, ferulic acid and essential oils are used as antibacterial materials. These compounds are natural substances with low toxicity and environmental safety. Through the structural optimization and the analysis of binary combined bacteriostatic efficiency, the MIC values of chlorobutyl ferulate (**2e**) and peppermint essential oil (EO$_1$) were 0.64 mg/mL and 2.02 mg/mL, respectively, and the MIC value of **2e**-EO$_1$ (mass ratio 1:1.5) was 0.40 mg/mL. The growth rate of bacteria treated with **2e**-EO$_1$ was inhibited, the OD$_{590nm}$ value of cell membrane decreased by 57.83%, and the expression levels of *hrpB*, *pehC*, *pilT*, *polA*, *aceE*, *egl*, and *phcA* were downregulated to 18.81%, 30.50%, 14.00%, 44.90%, 86.79%, 23.90%, and 27.56%, respectively. The results showed that **2e**-EO$_1$ had a synergistic inhibitory effect against *R. solanacearum*. It significantly affected the formation of the bacterial cell membrane and the expression of pathogenic genes. Consequently, **2e**-EO$_1$ provides the potential to become a green pesticide and can promote the sustainability of the agricultural ecological environment.

**Keywords:** *Ralstonia solanacearum*; ferulic acid; essential oil; antibacterial activity; synergistic effect; pathogenic gene

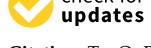



## 1. Introduction

The sustainability of agriculture is related to the development of the national economy and society's stability. By 2050, the crop output will be far lower than the needs of the growing population [1]. Plant diseases are the main factors affecting crop losses, reducing the yield by 14% [2]. It is considered that soil-borne diseases are more restrictive than seed-borne or air-borne diseases, accounting for 10–20% of the annual yield loss [3]. *Ralstonia solanacearum* is one of the ten most harmful plant bacteria and is distributed all over the world, especially in subtropical and tropical regions [4]. This soil-borne bacterium can infect more than 450 plant species of 54 botanical families [5] and cause huge direct economic losses every year [6]. The pathogenic genes of *R. solanacearum* have coevolved with the external environment and are characterized by multiple toxic factors [7]. Difficulties are associated with controlling this pathogen due to its ability to grow endophytically, its relationship with weeds, and its survival in soil [6]. Therefore, it is of great significance to research how to effectively inhibit *R. solanacearum* for agricultural sustainability.

Generally, a 1% increase in crop output per hectare is associated with a 1.8% increase in pesticide use per hectare [8]. However, it is not significant for many traditional agrochemicals against *R. solanacearum* [9]. Moreover, excessive chemical pesticides always bring organic residues and the production of resistant strains [10], which cause the unsustainability of the environment. Several effective fumigants are restricted due to their

serious environmental hazards, such as metam sodium and chloropicrin. It is reported that many plant residues derived from, e.g., chili [11], Chinese gall [12], and clove [13], have antimicrobial activities and the indirect suppression of pathogens through improved soil properties [14]. Biopesticides have emerged as a sustainable alternative leading to safe organic farming. At the global level, the environment-friendliness and target-specificity of biopesticides are gaining wide popularity [15].

Ferulic acid (**1**) and Essential oils (EO) are common natural metabolites with diverse bioactivities [16]. So far, there have been many reports on the effect of these substances and their analogs against *R. solanacearum*. Endophytes from the roots of *Solanum surattense* secrete a vast array of bioactive metabolites containing **1** to protect host plants against *R. solanacearum* [17]. Lansiumamide B, isolated from the seeds of *Clausena lansium*, the minimum inhibitory concentration (MIC) of which is 0.13 mg/mL, and the control efficiency against tobacco bacterial wilt is nearly 40 times higher than streptomycin [18]. EOs contain a variety of chemicals that have inhibitory activity against *R. solanacearum* [19]. Thyme oil, marjoram oil, and caraway oil have a significant antibacterial effect against R. solanacearum of tomato. Under field experiments, thyme oil treatment against tomato bacterial wilt reduces it by 94.8% [20]. Bacterial wilt of sweet pepper in a field treated with palmarosa oil, the latency period is increased (38%), the bacterial wilt index (36%), and the area under the disease progress curve (38%) are reduced [21]. In addition, due to low toxicity [22] and environmental friendliness [23], it is sustainable for **1** and EO against *R. solanacearum* compared with chemical pesticides.

Essentially, **1** belongs to the group of hydroxycinnamic acids (HCAs), and they are known to play multifunctional roles in rhizospheric plant-microbe interactions. In response to root pathogens, natural compounds that form chemical barriers play a key role in preventing pathogens from infecting plant roots, and many plants release de novo synthesized HCAs into the rhizosphere [24]. EO is a complex, volatile mixture that demonstrates antibacterial activity individually or as mixtures; its mechanism of bacteriostasis is very complicated and may be described as EO acting on membrane integrity [25]. The combined use of **1** and EO can improve their antimicrobial activity and may eliminate *L. monocytogenes* from mildly acid food products [26]. Although the combined effects of EO changed depending on the strain and the type of EO used, generally, the use of combinations increased the efficacy of EO [27].

In this study, ferulic acid ester (**2**) was synthesized in order to enhance biological activity, the combined antimicrobial efficiency of a binary mixture was determined, and the mechanism of bacteriostasis was discussed. This research was ready for the development of potential green antimicrobial agents, thereby reducing the pollution of chemical pesticides and promoting the sustainability of agriculture.

## 2. Materials and Methods

### 2.1. Materials and Bacterial

Peppermint EO (EO$_1$), Artemisia EO (EO$_2$), Citronella EO (EO$_3$), Chamomile EO (EO$_4$), Fennel EO (EO$_5$) and Patchouli EO (EO$_6$) were purchased from Jishui Lianxing Spice Oil Co. Ltd. (Jian, China). Other reagents were purchased from Aladdin Reagent (Shanghai) Co., Ltd. (Shanghai, China). The *R. solanacearum* of mulberry (PRJNA782242) was previously isolated by the lab members. It was cultured at 30 °C in a medium I (purified water with 0.1% of casamino acids, 0.5% of glucose, and 1% of tryptone). Medium II was prepared by adding 0.017 g of agar powder to 1mL Medium I. Bacterial suspensions of *R. solanacearum* were approximately $10^8$ CFU/mL at the optical density (OD) of 0.3–0.5 at 600 nm.

### 2.2. Gas Chromatography Analysis

The EO was analyzed by gas chromatography-mass spectrometry [28]. HP-INNOWax polyethylene glycol (30 m × 0.32 mm × 0.25 μm) was used. The carrier gas was helium, with a flow rate of 2.0 mL/min. The chromatographic analysis started at 40 °C and was maintained for 2 min. The temperature was raised to 240 °C at a rate of 5 °C/min and

maintained for 18 min. The injector temperature was 250 °C with a split ratio of 20:1. The temperatures of the ionization source and transfer lines were 230 °C and 250 °C, respectively. The ionic energy was 70 eV, and the mass range was 40–400 Da.

### 2.3. Synthesis of Ferulic Acid Ester

Based on this esterification method [29] with minor modifications and following the general procedure as shown in Scheme 1, in brief, **1** (5 mmol), 4-dimethylaminopyridine (0.25 mmol), and N, N′-dicyclohexylcarbodiimide (6 mmol) was dissolved in 20 mL of tetrahydrofuran, alcohol (30 mmol) was slowly added dropwise to the mixed solution. Then it was stirred for 12 h under nitrogen flow at room temperature. After filtration and concentration, the required product was purified by column chromatography (petroleum ether: ethyl acetate = 6:1), and their structures were characterized by FT–IR NMR and MS.

R(**2a**):$CH_3CH_2$; R(**2b**):$CH_3(CH_2)_2$; R(**2c**):$CH_3CH_2CH_3$; R(**2d**):$CH_3(CH_2)_3$;
R(**2e**): $CH_2Cl(CH_2)_3$; R(**2f**):$CH_3(CH_2)_4$; R(**2g**):$CH_3(CH_2)_6$

**Scheme 1.** Synthesis of **2a–2g**.

### 2.4. Determination of the Inhibition Rate (IR)

Based on this spectrophotometric method with slight modifications [30], briefly, 40 μL of bacterial suspension was added to 156 μL of Medium I and then replenished with different concentrations of 4 μL of the sample (**1**, **2a–2g** and $EO_{1-6}$). The resulting solution was incubated at 30 °C for 24 h. The control check group (CK) used 4 μL of ethanol. The IR was calculated with Equation (1), and the half-maximal effective concentration ($EC_{50}$) value was obtained from the calculation of the fitted curve.

$$IR(\%) = \left( 1 - \frac{TF_{Sample} - T0_{Sample}}{TF_{Blank} - T0_{Blank}} \right) \times 100\% \tag{1}$$

$T0_{sample}$ and $TF_{sample}$ indicate the absorbance values of the bacterial suspension before and after adding the sample, respectively. $T0_{blank}$ and $TF_{blank}$ indicate the absorbance values of the bacterial suspension before and after adding the control solution, respectively.

### 2.5. Determination of the MIC Values

Based on the serial microdilution method [31], briefly, 1 μL of 1% 2,3,5-Triphenyltetrazolium chloride (TTC) and 1 μL of bacteria suspension were inoculated into 198 μL of Medium I containing the samples (**2e** and $EO_{1-6}$, 0.12–9.6 mg/mL). The mixture was incubated at 30 °C for 24 h. The MIC value is defined as the minimum concentration of the sample that does not produce a pink color.

### 2.6. Determination of Combined Antimicrobial Efficiency

Based on this checkerboard method [32] with minor modifications, briefly, 1 μL of 1% TTC and 1 μL of bacteria suspension were inoculated into 198 μL of Medium I containing equal volumes of different concentrations of **2e** and $EO_{1-6}$ (2, 1, 0.5 and 0.25 times MIC). The mixture was incubated at 30 °C for 24 h. The fractional inhibitory concentration index (FIC) was calculated using Equation (2).

$$FIC = \frac{MIC_{mixA}}{MIC_A} + \frac{MIC_{mixB}}{MIC_B} \tag{2}$$

The $MIC_A$ and $MIC_B$ values represented the lowest concentrations of component A and component B when used alone, and the $MIC_{mixA}$ and $MIC_{mixB}$ values represented the lowest concentrations of A and B when combined.

### 2.7. Determination of the Minimum Bactericidal Concentration (MBC)

Based on this method [33] with minor modifications, briefly, the ethanol solution of the samples (**2e**, $EO_1$, and 2e-$EO_1$) was added separately to plates with freshly sterilized Medium II and the final concentration of the sample was 0.1–3.2 mg/mL. The plate was immediately placed on the super-clean table and cooled for half an hour. Then, 1 µL of bacterial suspension was dropped in the center of the culture dish. The MBC value is defined as concentration without bacterial growth after 96 h at 30 °C.

### 2.8. Growth Curve Assay

Based on this method [34] with modifications, briefly, the bacterial suspension was diluted by one-thousandth with Medium I, the samples (**2e**, $EO_1$, and 2e-$EO_1$) were added with a final concentration of 0.2 mg/mL, and the resulting solution was incubated in a shaking table at 30 °C. The optical density was read spectrophotometrically at 600 nm with a time interval of 4 h. The growth of *R. solanacearum* was observed by the change in the absorbance value.

### 2.9. Biofilm Assay

Bacteria form biofilms on a wide range of abiotic surfaces [35]; the polyvinylchloride (PVC) microtiter plate assay [31] was used for the quantification of the biofilm with minor modifications. Briefly, 40 µL of bacterial suspension was added to 160 µL of Medium I and then supplemented with the samples (**2e**, $EO_1$, and 2e-$EO_1$) at final concentrations of 0.1–1.0 mg/mL. The resulting solution was removed carefully after cultivating for 24 h. The residue was cleaned with 200 µL of purified water and was fixed for 15 min with 200 µL of methanol. Then, 220 µL of 0.1% crystal violet was mixed with the biofilms for 30 min and was removed. The floating color was washed twice with 200 µL of purified water and was removed. The residue was dried at 25 °C for 30 min. The crystal violet adsorbed on the biofilms was dissolved with 200 µL of 95% ethanol for 30 min, and the $OD_{590nm}$ value of the solution was measured.

### 2.10. Influence of Pathogenic Gene Expression

The influence of pathogenic gene expression was analyzed according to this early report [36]. Briefly, 0.8 mg of the samples (**2e**, $EO_1$, and 2e-$EO_1$) were added to 4 mL of the bacterial suspension. The mixture was incubated at 30 °C for 24 h and centrifuged at 12,000 rpm for 3 min to collect the precipitates. The total RNA extraction process was carried out according to the protocol modifications. Briefly, the total RNA was extracted by the Trizol reagent (Invitrogen, Waltham, MA, USA) and was purified by DNase I (Promega, Madison, WI, USA). The Prime Script RT Reagent Kit with gDNA Eraser (Takara Biotechnology, Otsu, Japan) was used to synthesize the cDNA, and the relative expression of pathogenicity-related genes (*hrpB*, *pehC*, *pilT*, *polA*, *aceE*, *egl* and *phcA*) was determined by the Light Cycler 96 real-time PCR system. Then, 20 mL of the reaction system was made up of 10 mL of TB Green, 6.4 mL of RNase-free double-distilled water, 2 mL of cDNA, 0.8 mL of the forward primer (F), and 0.8 mL of the reverse primer (R). The specific primers were synthesized as shown in Table 1, and the *16S rRNA* of *R. solanacearum* was used as CK.

**Table 1.** Primer information of the pathogenicity-related genes of *R.solanacearum*.

| Primer Name | Nucleotide Sequence(5′→3′) | Size (bp) |
|---|---|---|
| *hrpB* | F: TTCTCGATGATGTAGCGATAGG<br>R: GCTGGAATTTTCGACTTCCTCTA | 238 |
| *pehC* | F: GTTGTTCGGATTGCTGTACG<br>R: AGTCAAACGATTGCCTGAACTA | 227 |
| *pilT* | F: AAGAACAAAGCGTCTGATCTGC<br>R: CTTCCAGGTTTTCTTCGTAATGCT | 175 |
| *polA* | F: GGAATGTCGGAAAGTCAAGAAA<br>R: CTTGTAGGCGGGGTACAGTTC | 238 |
| *ace* | F: GCCTATGTGCGTGAGTTCTTCT<br>R: CTTCGAACTTGACGTACGGAAC | 338 |
| *egl* | F: CAGCGCGACCTACTACAAGA<br>R: TCATCAGCCCGAAGATGAC | 299 |
| *phcA* | F: GGACATGATCTTCACGGTCAACT<br>R: GACTCATCCTCCTTTTCTGCATC | 298 |
| *16S rRNA* | F: CTAGAGTGTGTCAGAGGGAGGTAGA<br>R: ATGTCAAGGGTAGGTAAGGTTTTTC | 349 |

*2.11. Statistical Analysis*

All tests were performed in triplicate, and the experimental data was recorded as the average value and stdev (SD). Variance analysis performed was one-way analysis and Tukey's test.

## 3. Results and Discussion

*3.1. Synthesis of Ferulic Acid Ester*

The yields of **2a–2g** were 74.57 ± 1.45%, 72.31 ± 2.38%, 69.75 ± 1.48%, 65.42 ± 2.49%, 67 ± 3.38%, 69 ± 3.29%, and 67 ± 3.36%, respectively. The spectral data of Figures S1, S2, and Figure 1 confirmed the chemical structure of **2a–2g**. In the ESI-MS analysis, 223, 237, 237, 251, 285, 265, and 293 were molecular ion peaks of **2a–2g**, respectively.

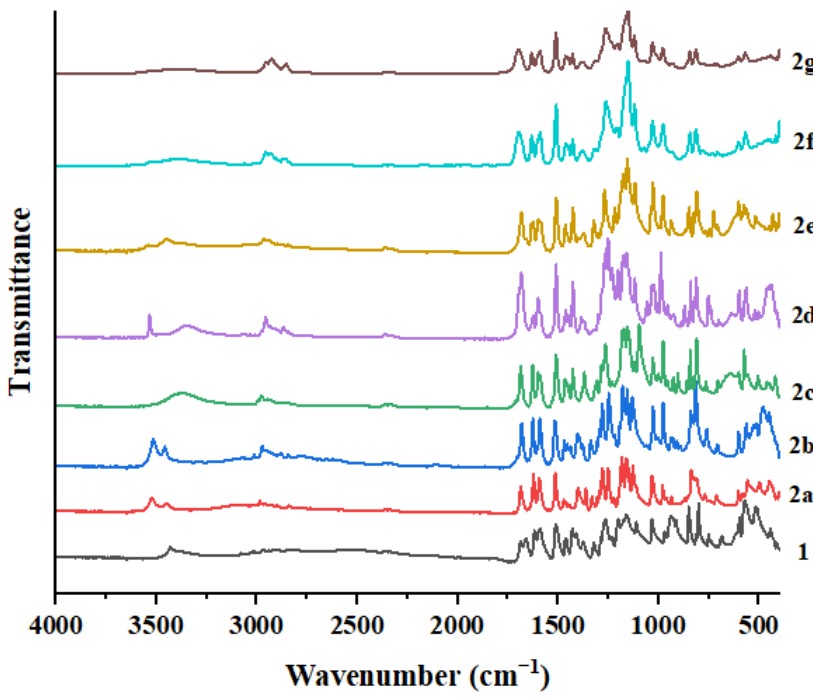

**Figure 1.** FT-IR spectrum of ferulate esters. **1**: Ferulic acid, **2a**: Ethyl ferulate, **2b** Propyl ferulate, **2c**: Isopropyl ferulate, **2d**: Butyl ferulate, **2e**: Chlorobutyl ferulate, **2f**: Amyl ferulate, **2g**: Heptyl ferulate.

The characteristic structure of this series is similar. **2e** is an example of the structural analysis of this series. For the $^1$H NMR analysis, $^1$H chemical shifts of the protons of the aromatic ring were assigned at δ 7.17, δ 7.05, and δ 6.79 ppm, and a shift signal at δ 4.15 ppm corresponded to the hydrogens of the first carbon in the ester alkyl chain, which suggested that the ester group was obtained. Besides, the other signals of chemical shifts between δ 1.68 and 0.93 δ ppm were assigned to the hydrogens of the other carbons in the alkyl chain.

For the FT–IR analysis, 3450 cm$^{-1}$ was the stretching vibration peak of –OH, 2970 cm$^{-1}$ was the stretching vibration peak of C–H, 1684 cm$^{-1}$ was the stretching vibration peak of C=O, 1634 cm$^{-1}$, 1624 cm$^{-1}$, 1600 cm$^{-1}$, and 1509 cm$^{-1}$ were the vibration peaks of conjugated double bonds of a benzene ring and its substituents, 1464–1427 cm$^{-1}$ were the bending vibration peaks of C–H, 1323–1269 cm$^{-1}$ were the C–O vibration peaks on a benzene ring, and the wide peak near 1156 cm$^{-1}$ was the C–O–C vibration peak of an ester.

### 3.2. Antimicrobial Activity of Ferulic Acid Ester against R. solanacearum

Figure 2 showed that the inhibition rate of **2** was proportional to the concentration, and the antibacterial activity was stronger than **1**. This was consistent with the report that electron-donating substituents on the benzene ring might enhance the antibacterial activity [37]. At the concentration of 0.32 mg / mL, the inhibition rates of **2a**, **2b**, **2c**, **2d**, **2e**, **2f**, and **2g** were 74.62 ± 2.42%, 79.26 ± 3.13%, 82.04 ± 4.41%, 84.13 ± 4.38%, 89.51 ± 4.84%, 72.13 ± 6.30% and 70.25 ± 4.83%, respectively. Their EC$_{50}$ values were calculated to be 0.12 ± 0.002 mg/mL, 0.12 ± 0.004 mg/mL, 0.12 ± 0.004 mg/mL, 0.10 ± 0.005 mg/mL, 0.07 ± 0.003 mg/mL, 0.13 ± 0.005 mg/mL, and 0.13 ± 0.002 mg/mL respectively. Therefore, the activity of short-chain ferulate varies little with different alkyl numbers, and the activity of butyl ester (**2d**) is slightly stronger than that of other short-chain alkyl esters (**2a**, **2b**, **2c**, **2f**, **2g**). Butyl chloride (**2e**) is larger than butyl ester (**2d**), which may be related to the bacteriostatic effect of chlorine [38].

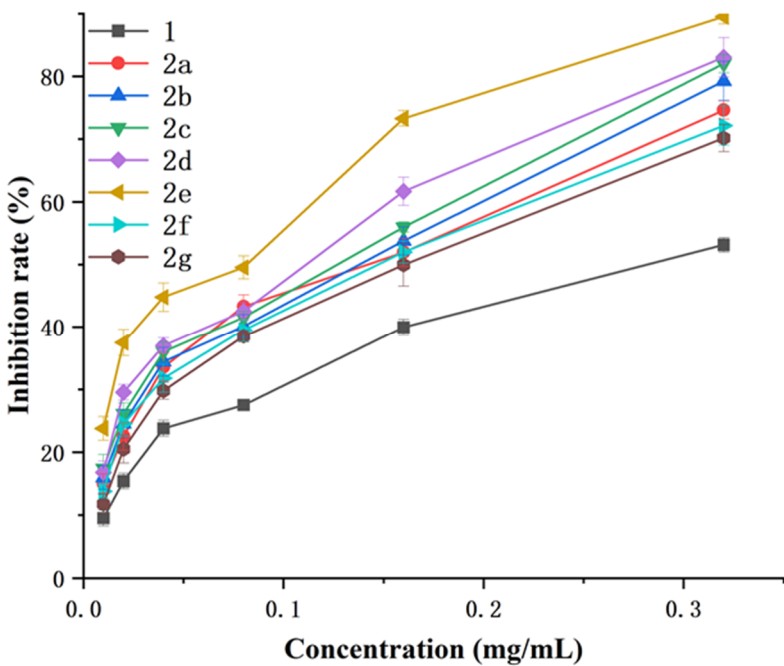

**Figure 2.** Antimicrobial activity of ferulic acid ester against *R. solanacearum*. **1**: Ferulic acid, **2a**: Ethyl ferulate, **2b** Propyl ferulate, **2c**: Isopropyl ferulate, **2d**: Butyl ferulate, **2e**: Chlorobutyl ferulate, **2f**: Amyl ferulate, **2g**: Heptyl ferulate. The OD$_{600nm}$ values of the bacterial solution was detemined after cultivation for 24 h at 30 °C. The EC$_{50}$ values were fitted by Logistic evaluation.

### 3.3. Combined Antimicrobial Efficiency of 2e and EO

Table 2 shows that the MIC value of **2e** against *R. solanacearum* was $0.64 \pm 0.03$ mg/mL. According to the FIC index, a mixture of **2e** and $EO_1$ showed a synergistic effect, and a mixture of **2e** and other EOs showed an additive effect. The bacteriostatic effect of **2e** and six EOs were stronger than that of either alone. This may be because of the multi-target effect and pharmacokinetic effects [39], and there may be synergy between the major and minor components of EO [40] and between EO and bacteriostatic agents [41]. Among six combinations, **2e**-$EO_1$ was the most significant; the MIC value of **2e**-$EO_1$ was $0.4 \pm 0.05$ mg/mL, and their optimal mass ratio of **2e** and $EO_1$ was 1:1.5. The amount of **2e** in the composition was reduced to 25%. Figure 3 showed that the MBC values of **2e**, $EO_1$, and **2e**-$EO_1$ were $1.60 \pm 0.08$ mg/mL, $3.20 \pm 0.14$ mg/mL, and $0.80 \pm 0.06$ mg/mL, respectively. The $EC_{50}$ value of **2e** $EO_1$ was $0.05 \pm 0.004$ mg/mL, as shown in Figure S3. The main component of $EO_1$ was menthol (65.38%), as shown in Table 3. These results verified that the bacteriostatic synergy of **2e**-$EO_1$ was significant. According to another report [42], $EO_1$ can increase membrane permeability and inhibition of bacterial quorum sensing ability in multidrug-resistant *E. coli*, aiding in the reversal of antibiotic resistance.

**Table 2.** FIC index of the combination of a mixture of **2e** and EO against *R. solanacearum*.

| mg/mL | 2e + EO$_1$ | | 2e + EO$_2$ | | 2e + EO$_3$ | | 2e + EO$_4$ | | 2e + EO$_5$ | | 2e + EO$_6$ | |
|---|---|---|---|---|---|---|---|---|---|---|---|---|
| MIC | $0.64 \pm 0.03$ | $2.02 \pm 0.08$ | $0.64 \pm 0.03$ | $2.42 \pm 0.09$ | $0.64 \pm 0.03$ | $1.20 \pm 0.08$ | $0.64 \pm 0.03$ | $4.51 \pm 0.09$ | $0.64 \pm 0.03$ | $4.44 \pm 0.07$ | $0.64 \pm 0.03$ | $4.38 \pm 0.05$ |
| MIC$_{mix}$ | $0.16 \pm 0.02$ | $0.24 \pm 0.03$ | $0.32 \pm 0.04$ | $0.32 \pm 0.04$ | $0.32 \pm 0.06$ | $0.13 \pm 0.04$ | $0.32 \pm 0.04$ | $1.32 \pm 0.04$ | $0.32 \pm 0.04$ | $2.23 \pm 0.07$ | $0.32 \pm 0.04$ | $2.25 \pm 0.05$ |
| FIC | $0.37 \pm 0.09$ | | $0.63 \pm 0.06$ | | $0.61 \pm 0.23$ | | $0.79 \pm 0.18$ | | $1.00 \pm 0.07$ | | $1.01 \pm 0.17$ | |
| Effect | S | | AD | | AD | | AD | | AD | | AD | |

**2e**: Chlorobutyl ferulate, $EO_1$: Peppermint EO, $EO_2$: Artemisia EO, $EO_3$: Citronella EO, $EO_4$: Chamomile EO, $EO_5$: Fennel EO, $EO_6$: Patchouli EO. The FIC $\leq 0.5$ indicated a synergistic effect (S). $0.5 <$ FIC $\leq 1$ indicated additive effect (AD), $1 <$ FIC $\leq 4$ indicated no interactive effect (NI), and FIC $> 4$ indicated antagonistic effect (A).

**Table 3.** Relative percentages of the main constituents of $EO_1$ determined by the GC analysis.

| Main Components | Retention Time (min) | Retention Indices | | Percentages of the Main Constituents (%) |
|---|---|---|---|---|
| | | SI | RSI | |
| (+)-Dipentene | 15.97 | 874 | 877 | 3.00 |
| L-menthone | 20.13 | 942 | 958 | 8.07 |
| Menthone | 20.49 | 858 | 865 | 5.44 |
| Menthol | 20.76 | 947 | 952 | 65.38 |
| Isomenthol acetate | 24.09 | 946 | 954 | 2.65 |

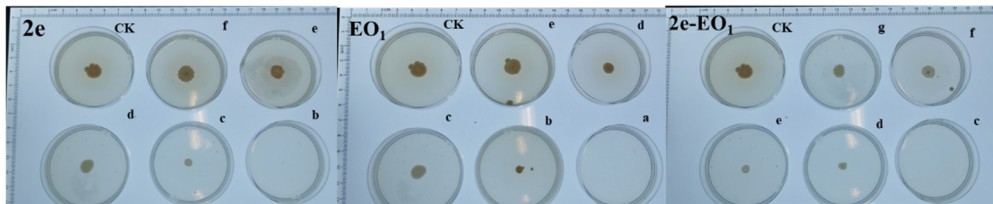

**Figure 3.** Bactericidal activity of **2e**-$EO_1$. CK: Ethanol, **2e**: Chlorobutyl ferulate, $EO_1$: Peppermint EO, and **2e**-$EO_1$: Mixture of chlorobutyl ferulate and peppermint EO (1:1.5). a. b, c, d, e, f and g were 3.2, 1.6, 0.8, 0.4, 0.2, 0.1, and 0.05 mg/mL, respectively. The culture condition was 30 °C for 96 h.

### 3.4. The Growth Curve of R. solanacearum

The effect of **2e**-$EO_1$ on the growth curve of *R. solanacearum* is shown in Figure 4. The growth process of CK groups showed that the lag phase of *R. solanacearum* was 0–8 h, where there was a lesser growth of bacteria, and the log phase was 8–24 h, where the growth rate was rapid. This result was consistent with another report [43]. In the growth process of **2e**, $EO_1$, and **2e**-$EO_1$, the absorbance value of *R. solanacearum* changed slowly at 0–12 h, only increased by $0.027 \pm 0.013$, $0.045 \pm 0.016$ and $0.023 \pm 0.009$ respectively, and their growth rate was significantly slower than that of the control group ($0.127 \pm 0.02$). Thus, **2e**, $EO_1$, and **2e**-$EO_1$ have good antimicrobial effects in the first 12 h. The absorbance value of *R.*

*solanacearum* treated with **2e**, $EO_1$, and **2e**-$EO_1$ increased significantly from 12 to 24 h, were $0.153 \pm 0.006$, $0.292 \pm 0.005$, and $0.126 \pm 0.004$, respectively. Among them, the change in the **2e**-$EO_1$ treatment group was relatively little, and the antimicrobial effect was stronger than that of the **2e** and $EO_1$ treatment groups.

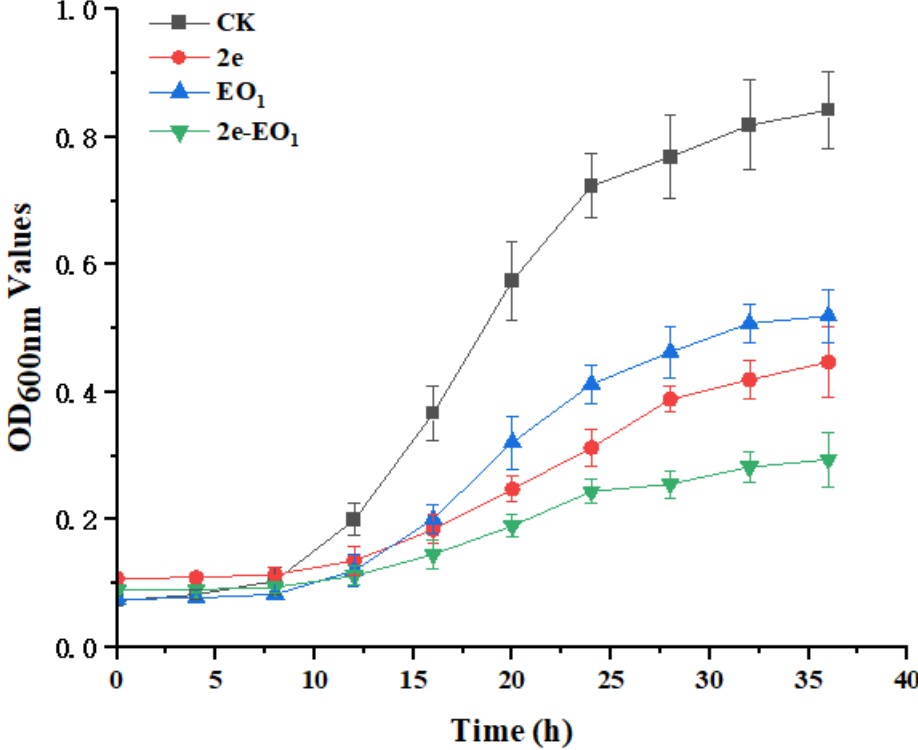

**Figure 4.** The growth curve of R. solanacearum treated with **2e**-$EO_1$. CK: Ethanol, **2e**: Chlorobutyl ferulate, $EO_1$: Peppermint EO, and **2e**-$EO_1$: Mixture of chlorobutyl ferulate and peppermint EO (1:1.5). The culture condition was 30 °C for 36 h. The density of *R. solanacearum* ($OD_{600nm} \approx 1.0$) was approximately $10^9$ CFU/mL.

### 3.5. Biofilm Assay of R. solanacearum

Figure 5 showed that **2e**, $EO_1$, and **2e**-$EO_1$ had different degrees of inhibition for the biofilm formation of *R. solanacearum*, and **2e**-$EO_1$ was especially significant. The $OD_{590nm}$ value of bacteria treated with 1.0 mg/mL of **2e**-$EO_1$ was $0.35 \pm 0.03$, and the $OD_{590nm}$ value was decreased by $57.83 \pm 6.46\%$. Similar to many plant pathogens, bacterial biofilms formed in the roots of host plants can effectively colonize the xylem walls of host plants [44] and promote bacterial invasion and infection [45]. According to another report [46], the specific phenolic exudates in plants infected with *R. solanacearum* that exhibit promote plant resistance against pathogens and antibacterial activity. In addition, the mechanism of action of EO against *R. solanacearum* might be described as acting on membrane integrity [47]. The reason was that the membrane structure of the pathogen was damaged, resulting in the thinning of the cell membrane and irregular cavities in cells.

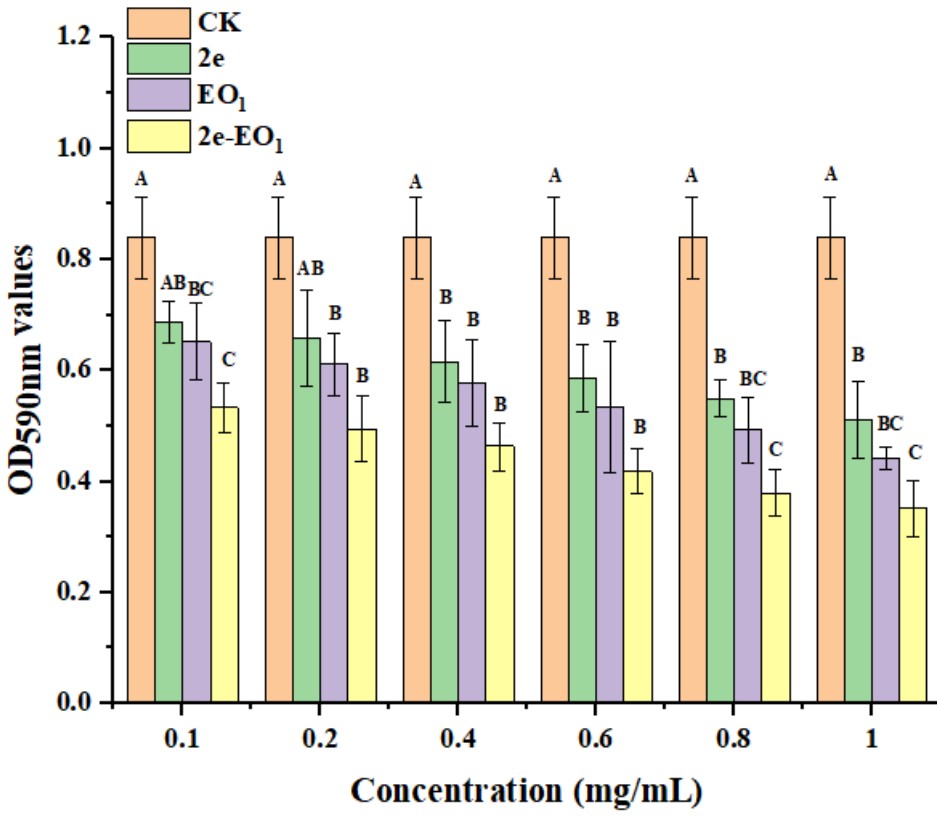

**Figure 5.** The biofilm assay of *R. solanacearum* treated with **2e**-EO$_1$. CK: Ethanol, **2e**: Chlorobutyl ferulate, EO$_1$: Peppermint EO, **2e**-EO$_1$: Mixture of chlorobutyl ferulate and peppermint EO (1:1.5). Biofilm assay after treatment with different concentrations of **2e**, EO$_1$ and **2e**-EO$_1$ at 30 °C for 24 h in the 96-well plate. The letters (A, B, C) on the column indicate the differences between the different treatment groups; the same letters indicate that the difference is not significant ($p > 0.05$), and completely different letters indicate significant differences ($p < 0.05$).

### 3.6. Influence of Pathogenic Gene Expression

Figure 6 shows the expression of these pathogenic genes of *R. solanacearum* after treatment with **2e**, EO$_1$, and 2e-EO$_1$. Compared with CK, the expression levels of *pilT*, *polA*, *aceE*, *egl*, and *phcA* treated with **2e** were downregulated to 47.00 ± 9.90%, 89.78 ± 7.83%, 93.17 ± 9.19%, 85.86 ± 8.42% and 54.48 ± 3.92%, respectively. The expression levels of *pilT*, *polA*, *aceE*, and *phcA* treated with EO$_1$ were downregulated to 23.53 ± 2.41%, 75.44 ± 7.03%, 26.30 ± 3.83% and 12.81% ± 1.56%, respectively. The expression levels of *hrpB*, *pehC*, *pilT*, *polA*, *aceE*, *egl*, and *phcA* treated with **2e**-EO$_1$ were downregulated to 18.81 ± 4.79%, 30.50 ± 4.03%, 14.00 ± 1.97%, 44.90 ± 8.00%, 86.79 ± 12.48%, 23.90 ± 2.01%, and 27.56 ± 4.14%, respectively. Among these pathogenic genes, *hrpB* is a core type II and type III secretion system regulator gene [48]. *pehC*, *pilT*, *polA*, and *aceE* are important factors in the early stage of bacterial blight, and *egl*, along with *phcA*, are important factors in the late stage of bacterial blight [21]. This result indicated that the drug has a multi-level inhibitory effect on the infection of *R. solanacearum*. Under the synergistic effect of **2e** and EO$_1$, the target of action was expanded. The multi-target mechanism is beneficial in reducing the drug resistance of bacteria.

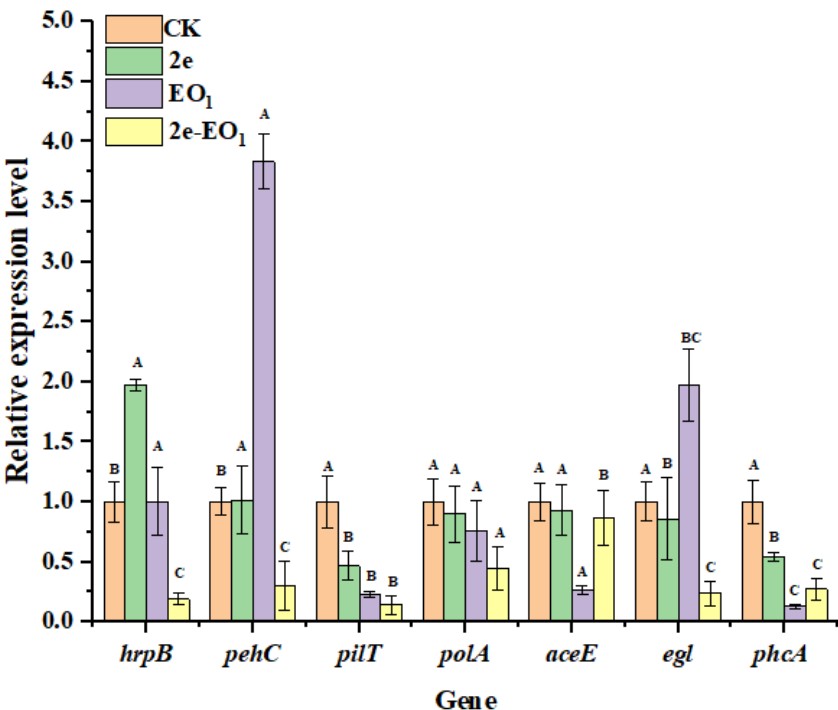

**Figure 6.** Effect of the expression of the pathogenicity-related gene of *R. solanacearum* treated with **2e**-$EO_1$. CK: Ethanol, **2e**: Chlorobutyl ferulate, $EO_1$: Peppermint EO, **2e**-$EO_1$: Mixture of chlorobutyl ferulate and peppermint EO (1:1.5). **2e**, $EO_1$ and **2e**-$EO_1$ were added to the bacterial suspension ($OD_{600nm} \approx 1.0$) and incubated overnight at 30 °C for 24 h. The letters (A, B, C) on the column indicate the differences between the different treatment groups; the same letters indicate that the difference is not significant ($p > 0.05$), and completely different letters indicate significant differences ($p < 0.05$).

## 4. Conclusions

In this research, ferulic acid esters (**2a–2g**) were synthesized, the inhibitory activity against *R. solanacearum* was determined, and the antibacterial combination of **2** and EO was optimized. The antibacterial curve of the composition and the expression of pathogenic factors were discussed. This paper showed that the $EC_{50}$ value of chlorobutyl ferulate (**2e**) was 0.07 mg/mL, and the composition of **2e** and $EO_1$ had a synergistic effect. The MIC and MBC values of **2e**-$EO_1$ were 0.40 mg/mL and 0.80 mg/mL, respectively. In the growth curve of *R. solanacearum*, the **2e**-$EO_1$ treatment group grew slowly, and the multiple pathogenic genes (*hrpB*, *pehC*, *pilT*, *polA*, *aceE*, *egl*, and *phcA*) of *R. solanacearum* treated with **2e**-$EO_1$ were significantly downregulated. These results show that **2e**-$EO_1$ has significant bacteriostatic properties against *R. solanacearum*. In addition, **2** and EO are natural substances with low toxicity and environmental safety, and **2e**-$EO_1$ has the potential to become a green pesticide.

**Supplementary Materials:** The following supporting information can be downloaded at: https://www.mdpi.com/article/10.3390/su142416348/s1, Figure S1. ESI-MS analysis of ferulate esters; Figure S2. 1H NMR spectrum of ferulate esters; Figure S3. Antimicrobial activity of **2e**-$EO_1$ against *R. solanacearum*.

**Author Contributions:** Conceptualization, F.-A.W.; data curation, Q.-B.T.; formal analysis, Q.-B.T. and S.S.; funding acquisition, Q.-B.T. and F.-A.W.; investigation, H.-C.S.; methodology, Q.-B.T. and P.L.; project administration, F.-A.W.; resources, F.-A.W.; supervision, F.-A.W.; validation, H.-C.S.; writing—original draft preparation, Q.-B.T.; writing—review and editing, Q.-B.T. and F.-A.W. All authors have read and agreed to the published version of the manuscript.

**Funding:** This study was supported by the Science and Technology Plan Project of Taizhou City (TN202124), the Key Research and Development Program (Modern Agriculture) of Zhenjiang City (NY2020005), the Key Research and Development Program (Modern Agriculture) of Jiangsu Province

(BE2021344), the Jiangsu Graduate Research Innovation Program (KYCX18_2347), the Innovation and entrepreneurship training program for college students of Jiangsu Province (201913981010Y), and the Special Fund for China Agriculture Research System (CARS-18).

**Institutional Review Board Statement:** Not applicable.

**Informed Consent Statement:** Not applicable.

**Data Availability Statement:** The data and contributions presented in the study are included in the article. Further inquiries can be directed to the corresponding author.

**Conflicts of Interest:** The authors declare no conflict of interest.

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
