# Peer review of "Antibacterial Activity of Ferulic Acid Ester against Ralstonia solanacearum and Its Synergy with Essential Oils"

_sustainability, doi:10.3390/su142416348_

Round 1
Reviewer 1 Report
This study evaluates the effect of 7 ferulic acid ester (FAE) derivates and 6 different essential oils (EOs), alone or in certain combinations, on the growth of the plant pathogenic bacterium Ralstonia solanacearum in vitro. There are certain aspects of the study that are interesting, including the differential efficacy of different FAEs and the synergism of one specific FAE-EO combination.
Having said this, I do not consider the manuscript acceptable for publication in Sustainability. This is based on the following criticisms:
1) The manuscript is outside the scope of the journal. Sustainability publishes on the “technical, environmental, cultural, economic and social sustainability of human beings, which provides an advanced forum for studies related to sustainability and sustainable development.” The current study is a conventional laboratory microbiology study that does not directly address sustainability. The authors argue in the Introduction that their antimicrobial agents might be more sustainable than conventional agrichemicals, but that remains to be determined.
2) Consisting exclusively of in vitro experiments, the study is too preliminary to warrant publication at this stage. Additional experiments in more realistic conditions are needed to document shelf life of the compounds and their efficacy in the greenhouse or field.
3) The methodology section is too cryptic and does not contain sufficient information to reproduce the work. For all experiments, the overall experimental design, number of biological and technical replicates, and statistical analysis of the data should be described. In many places, “the sample” is mentioned, but it is not clear what compound(s) or concentrations are tested in each experiment. Section 2.8 is labeled as “cell membrane formation”, but it seems that actually biofilm formation (a very different process) is being measured. The methodology for the pathogenicity gene expression study is lacking in section 2.9, including an explanation of which genes were tested, why (i.e., biological rationale) and how.
4) The English writing of the manuscript (grammar and structure) needs to be improved. The authors should utilize a reputable English language editing service prior to considering resubmission.
Additional comments (by line number):
37-40: The potential environmental harm associated with the use of agrichemicals against the pathogen is exaggerated. This section should focus on the lack of efficacy and risk of resistance development instead.
48: Salicylic acid (aspirin) is not particularly antibacterial. I would delete this statement.
95: What was the “sample” here? Each of compounds 2a through 2g? Also describe experimental design, number of biological and technical replicates, and statistical analysis of the data. This is also needed for all the other experiments described in Materials & Methods.
103: What is TTC?
107 and 116: There are two sections labeled as 2.5.
116: Shouldn’t this section (“Gas chromatography analysis”) be in the beginning or end of Materials & Methods, rather than between the different growth inhibition assays?
110: What is “2” here in the context of the previously mentioned compounds 2a through 2g? Which compound exactly was used?
115: Explain what “a” and “b” are in this context. I assume they are not the same as 2a and 2b.
124-136: Why are the MBC and growth curve assay sections not presented after MIC?
138: In one sentence, explain the principle of the cell membrane formation assay here. In other words, how is what is measured related to cell membrane formation? Line 244 seems to suggest that biofilm formation was measured, which is very different from “cell membrane formation”.
154: What were the pathogenicity genes analyzed here, and how was this done (methodology)?
205: According to the FIC index, what is the cutoff between additive and synergistic?
259: What is the “drug” in this context?
Fig. 2: The legend should mention what 1 and 2a through 2g are (similar to Fig. 1). In addition, briefly mention the experimental conditions (media type used, incubation time).
Table 1: The footnote should explain what 2e and EO1 through EO6 are.
Fig. 3: The legend should mention what 2e and EO1 are. In addition, briefly mention the experimental conditions (media type used, incubation time).
Fig. 4: The legend should mention what 2e and EO1 are. In addition, briefly mention the experimental conditions (media type used, incubation time). Mention the sample size and whether error bars are SD or SE. In the legend or somewhere in the text it should be stated what an OD600 of 0.8 corresponds to (in terms of bacterial cell density).
Fig. 5: The legend should mention what 2e and EO1 are. In addition, briefly mention the experimental conditions (media type used, incubation time). Mention the sample size and whether error bars are SD or SE.
Reviewer 2 Report
The manuscript "Antibacterial activity of ferulic acid ester against Ralstonia solanacearum and its synergy with essential oils" presents interesting research results, but requires minor corrections.
Detailed comments:
line 73-74 - Have the EOs been characterized by the manufacturer?
line 76-80 - how will they identify the microorganisms? Has the strain been deposited in the public collection?
line 159-161 - please describe in more detail the methods of statistical analysis used in the research.
Chapter 3.3 - the chapter requires editing, so as not to repeat all the results obtained in the text, since they are in the table, I propose to leave only the most important ones and without specifying the standard deviation, which is in the table.
Figure 5 - statistical analysis is missing, it is best to use one-way analysis and Tukey's test.
Figure 6 - also no statistical analysis.
Reviewer 4 Report
Please see the PDF file with comments
